# Effect of Sowing Date on the Growth Characteristics and Yield of Growth-Constrained Direct-Seeding Rice

**DOI:** 10.3390/plants12091899

**Published:** 2023-05-06

**Authors:** Rong-Ping Zhang, Ning-Ning Zhou, Ri-Gui Ashen, Lin Zhou, Ting-Yu Feng, Ke-Yuan Zhang, Xue-Huan Liao, Li-Se Aer, Jian-Chao Shu, Xue-Wu He, Fei Gao, Peng Ma

**Affiliations:** 1School of Life Science and Engineering, Southwest University of Science and Technology, Mianyang 621010, China; zhzhrrpp@163.com (R.-P.Z.); zhouningning@mails.swust.edu.cn (N.-N.Z.); ashenxiaowang@163.com (R.-G.A.); zhoulin@mails.swust.edu.cn (L.Z.); fengty@mails.swust.edu.cn (T.-Y.F.); ky010302@163.com (K.-Y.Z.); lxh000818@163.com (X.-H.L.); rm88886666@163.com (L.-S.A.); 2Sichuan Seed Industry Co., Ltd., Chengdu 610000, China; 13258313222@163.com; 3Sichuan Taiwo Seed Industry Co., Ltd., Jiangyou 621700, China; hxwsctw123321@163.com (X.-W.H.); 13402357888@163.com (F.G.)

**Keywords:** growth-constrained direct-seeding rice, *Oryza sativa* L., sowing date, growth, population quality, yield

## Abstract

To investigate changes in the yield and physiological characteristics of indica hybrid rice varieties sown on different dates, we evaluated appropriate hybrid rice varieties and their optimal sowing dates in the hilly areas of Sichuan. Three popular indica rice varieties were used as experimental materials, and five sowing dates were set uniformly locally [16 May (SD1), 23 May (SD2), 30 May (SD3), 6 June (SD4), and 13 June (SD5)] to investigate differences in the yield characteristics, growth period, and dry matter accumulation. The results showed that, over the two years, the sowing-to-heading period and overall growth period of the three varieties shortened as the sowing date was delayed, and the difference in yield between the SD1 and SD2 treatments was not significant, owing to higher material accumulation after flowering and higher assimilative material transport capacity. These varieties are both photosensitive and tolerant to low temperatures. Among the three varieties tested, the Huangyouyuehesimiao (V3) cultivar had the highest yield, with 10.75 t ha^−1^ under the SD2 treatment. The impact of shifting the sowing date on yield components varied. Delaying the sowing date increased and then decreased the number of effective panicles, and the number of grains per panicle and the seed setting rate decreased by differing degrees. In summary, a high yield of indica hybrid rice can be maintained by sowing between 16 and 23 May each year in the study area. It indicated that indica hybrid rice in the hilly rice-producing region of Sichuan is highly adaptable to different sowing dates.

## 1. Introduction

Chinese methods of rice planting are critical for improving global food security [1,2,3]. Sichuan is one of China’s most important rice-producing provinces. Wheat–rice, rapeseed–rice, and green manure–rice are the most common cropping rotations in the hilly area east of Sichuan [4,5]. The rice-production season is limited due to the multiple-cropping conditions. The shorter the time between successive cropping, the better. To increase yield, the rice growth period should be extended as far as possible. Because of the contradiction between labor scarcity and high labor consumption during the stubble changing season, simplification of rice cultivation, planting, and other operations is needed to reduce labor consumption. The simpler the rice production mode, the better. Direct-seeded rice has become a main planting method, replacing transplanted rice. Direct seeding can save up to 25% on labor costs compared to transplanting seedlings [6]. Rice is the staple grain. Rice production should be short, simplified, and labor-saving in order to ensure grain ration supply. However, production should be steadily increased [7,8,9]. 

Temperature and light resources in the lower reaches of the Yangtze River are abundant in one season but scarce in two others, with regional differences [9,10,11]. Varieties of direct-seeded rice differ in terms of temperature and light sensitivity and growth characteristics, resulting in varying adaptability to temperature and light during the late season [12,13,14]. The appropriate sowing date of the different varieties may fundamentally affect their characteristics. 

Due to the irregular distribution of rainfall in spring and summer in the hilly area of northeast Sichuan, drought often occurs in the early stage of rice growth. The temporal and spatial distribution of rainfall does not coincide with the growth and development of rice and may lead to a seasonal drought [15].

Furthermore, due to the short stubble period of double-cropping rice in the hilly area of northeast Sichuan, the early stages of sowing preparation work can be delayed or missed, such as stubble cutting, tilling, harrowing, and leveling the field. Consequently, grain seeds are not planted in time. All previous stubble is returned to the field, and extensive or semi-extensive soil preparations are performed over a short agricultural consumption period, resulting in a variety of adverse seed and seedling conditions [16,17]. This significantly restricts the normal emergence and growth of direct-seeding rice, resulting in a low and unstable rice yield. We, therefore, refer to this as growth-constrained direct-seeding rice. This type of growth-constrained direct-seeding rice is produced extensively, limiting balanced growth and yield in large areas.

Systematic research on the growth and development characteristics of growth-constrained direct-seeding rice, the mechanism of coordinated stable and high yield, and the cultivation techniques, particularly the impact of appropriate sowing date, remains lacking. In this study, three widely used growth-constrained direct-seeding rice varieties were used to investigate sowing dates in 2021 and 2022. We studied the growth and yield formation characteristics of growth-constrained direct-seeding rice using different sowing date treatments to determine the appropriate sowing date and provide a reference for obtaining stable yields and improving the efficiency of modern direct-seeding rice in large areas, provide a sound theoretical basis for its high-yield and high-quality cultivation.

## 2. Results

### 2.1. Effects of Sowing Date on the Growth Process of Growth-Constrained Direct-Seeding Rice

Table 1 shows that the year, variety, sowing date, and their interactions have significant effects on the sowing-to-heading, heading-to-maturity, and total growth period of hybrid rice. The total growth period of the varieties tested in 2022 was shorter than that in 2021, mainly because the continuous high temperature before and after heading accelerated the growth process (Figure 1). The average whole growth period of V1 was 137.5 days, V2 was 134 days, and V3 was 125 days. There was no significant difference between V1 and V2; however, there was a significant difference between V1 and V3. The whole growth period for V1 was higher than that of V2 and V3 by 2.61% and 10.00%, respectively. The average sowing-to-heading period was shortened by 8.33 days, the heading-to-maturity period by 12.67 days, and the total growth period by 21.00 days.

The sowing-to-heading and total growth periods of the three varieties were shortened as the sowing date was delayed by two years. This indicated that the sowing date and temperature had a significant influence on growth-constrained direct-seeding rice.

### 2.2. Effects of the Sowing Date on Effective Accumulated and Average Daily Temperature of Hybrid Rice

The effective accumulated and average daily temperatures were significantly affected by year, sowing date, variety, and their interaction (Table 2). In general, by delaying the sowing date in 2021, the effective accumulated temperature of sowing-to-heading first increased and then decreased for V2 and V3 and was highest with the SD1 treatment for V1. The effective accumulated temperature of the tested varieties at heading-to-maturity also decreased significantly. In 2022, the effective accumulated temperature of V3 decreased trend, while it first increased and then decreased for the other varieties. Delaying the sowing date reduced the effective accumulated temperature of the total growth period for the tested varieties. The effective accumulated temperatures of subplots SD2, SD3, SD4, and SD5 were reduced by 0–3.15%, 1.10–5.56%, 2.68–9.96%, and 5.93–14.27%, respectively, compared with that of SD1. The sowing-to-heading EAT for V1 was 11.91 and 18.00% higher for V2 and V3, respectively. The EAT of the heading-to-maturity stage for V3 was 30.03% and 1.32% higher for V1 and V2, respectively. The whole growth period EAT was 1.41% and 4.72% higher for V2 and V3, respectively, compared to that of V1. The sowing-to-heading ADT was 0.57% higher for V1 compared to V2 and V3, respectively. The heading-to-maturity ADT was 12.13% and 5.10% higher for V1 and V2, respectively, compared to that of V3. The whole growth period ADT was 2.90% and 2.05% for V1 and V2 compared to that of V3, respectively.

The average daily temperature of the V1 sowing-to-heading period in 2022 increased and decreased as the sowing date was delayed, peaking with the SD4 treatment. The average daily temperature of the sowing-to-heading period for the other varieties increased as the sowing date was delayed, peaking with the SD5 treatment. The average daily temperature from heading-to-maturity decreased significantly as the sowing date was delayed. In 2021, the average daily temperature for the total growth period of the tested varieties showed a downward trend as the sowing date was delayed. In 2022, however, the average daily temperature of the total growth period first increased and then decreased as the sowing date was delayed, peaking with the SD2 treatment. For the tested varieties, the average effective accumulated temperature of the sowing-to-pumping period for the different sowing date treatments in 2022 was 5.41% lower than that in 2021. The effective accumulated temperature for the heading-to-maturity period in 2022 was 19.13% lower than that in 2021, and the effective accumulated temperature of the total growth period was 9.58% lower than that in 2021. The sowing-to-pumping period of the tested varieties in 2022 for different sowing date treatments was 2.58% higher than that in 2021; the average daily temperature of the pumping-to-ripening period was 3.48% higher than that in 2021; and the average daily temperature of the total growth period was 3.47% higher than that in 2021. The interaction between the sowing date and year had a substantial impact on the effective accumulated and daily average temperatures of growth-restricted direct-seeding rice.

### 2.3. Effects of the Sowing Date on the Leaf Area Index of Hybrid Rice in the Main Growth Stages

The leaf area index of growth-constrained direct-seeding rice at the maximum tillering and full heading stages is shown in Figure 2. The leaf area index at the maximum tillering stage increased and then decreased as the sowing date was delayed (Figure 2). With SD3 treatment, the leaf area index differed notably among varieties. The leaf area index of V2 was higher than that of V1 and V3. At the full heading stage, the leaf area index decreased, with the highest index obtained with the SD1 treatment. The leaf area index of the SD2, SD3, SD4, and SD5 treatments was compared with that of the SD1 treatment, with increases of 0.61–13.42%, 3.64–22.48%, 16.29–30.13%, and 25.04–39.39%, respectively. The full heading stage for V1 was 5.92% and 2.74% higher for V2 and V3, respectively.

### 2.4. Effects of the Sowing Date on Chlorophyll Content in Flag Leaves of Growth-Constrained Direct-Seeded Rice after Full Heading

The chlorophyll content of the tested varieties was the highest 0 days after full heading for each treatment (Figure 3). The chlorophyll content of the flag leaf decreased as growth progressed. The chlorophyll content of the flag leaf was highest with the SD3 treatment across all sowing dates. The chlorophyll content of flag leaf in SD1, SD2, SD4, and SD5 treatments was 11.50–2.29%, 10.86–26.55%, 4.79–6.92%, and 6.54–23.14%, respectively, highest than in the SD3 treatment at 21 days after full heading. This showed that delayed sowing resulted in delayed chloroplast degradation and improved the capacity for photosynthetic material production. At 0 days after full heading, the chlorophyll content of V2 in SD1 was 26.32% higher than that of V1 and 0.60% higher than that of V3. The chlorophyll content of V3 in SD2, SD3, SD4, and SD5 treatments was 40.21%, 19.58%, 23.83%, and 18.02%, respectively, higher than that of V1, and 11.30%, 11.63%, 10.81%, and 5.70%, respectively, higher than that of V2. Figure 3 shows that at 0 days of the full heading, the peroxidase activity of the tested varieties was the highest for each treatment and showed a decrease as growth proceeded. The peroxidase activity of the tested varieties was highest with the SD3 treatment among the different sowing dates. The peroxidase activities of the tested varieties were 36.17–57.58%, 21.21–25.00%, 0–21.21%, and 7.50–36.36% higher with the SD1, SD2, SD4, and SD5 treatments, respectively, compared to that with the SD3 treatment 21 days after full heading. By delaying the increase in peroxidase activity at the sowing date, the senescence of rice leaves could be delayed, thus maintaining a higher photosynthetic function of the leaves. The peroxidase activity of V2 seedlings in SD1 was 80.00% higher than that of V1 and 12.50% higher than that of V3 seedlings 0 days after full heading. The peroxidase activities of V3 in SD2, SD3, SD4, and SD5 were 14.93%, 34.26%, 83.10%, and 50.00% higher than those of V1, and 16.67%, 12.40%, 31.31%, and 20.83% higher than those of V2 for the same treatments.

### 2.5. Effects of the Sowing Date on Above-Ground Dry Matter Accumulation of Hybrid Rice during the Main Growth Period

Analysis of variance results for the dry matter accumulation in rice is presented in Table 3. The dry matter accumulation of direct-seeding rice varieties in each growth period was significantly affected by Y, C, SD, Y × C, Y × SD, and Y × C × SM. The dry matter accumulation of rice at the full-heading and mature stages was the highest in 2021. The dry matter accumulation of rice at the full-heading-to-mature stage and panicle was 23.5% and 10.41% higher in 2022 than that in 2021, respectively (Table 4). Compared with V3, dry matter transport in V1 and V2 increased by 43.29% and 9.67%, respectively, from the full heading to maturity stage in the different varieties. Compared with V3, the dry matter accumulation in V1 and V2 increased by 14.93% and 8.43%, respectively, in the ears of different varieties. In 2021, the dry matter accumulation of direct-seeding rice varieties in each growth period decreased as the sowing period was delayed. In 2022, the dry matter of V1 in each growth period first increased, then decreased as the sowing date was delayed, with the greatest increase obtained with the SD2 treatment. The dry matter of other varieties decreased as the sowing date was delayed, consistent with the trend observed in 2021. The maturity stage panicle for V3 was higher by 12.26% and 10.36% for V1 and V2, respectively. The heading-to-maturity for V3 was higher by 43.27% and 9.61% for V1 and V2, respectively. The dry matter increase of panicle for V3 was higher by 14.95% and 8.43% for V1 and V2, respectively. The results of the two-year experiment indicated that in V3, the highest dry matter accumulation in the ears occurred with the SD3 sowing date treatment in 2022 and was 9.93 t ha^−1^. An appropriate sowing date led to the accumulation and transportation of dry matter in rice, whereas a late sowing date was not conducive to increasing the rice yield.

### 2.6. Yield and Yield Components

Analysis of variance results for the yield and yield components are presented in Table 4. Grain yield and yield components were significantly affected by Y, C, SD, Y × SD, and Y × C × SD. There were significant differences in grain yield and yield components among years and cultivars as the sowing date was delayed (Table 4). Panicles for the SD3 treatment were the highest among the five sowing dates in 2021, and those from the SD4 treatment were the highest among the five sowing dates in 2022. This was because more productive tillers formed due to high temperatures, and ineffective tilling was reduced. The spikelets per panicle for the SD1 treatment were the highest among the five sowing dates in 2021 but were higher for the cultivar V1 in the SD2 treatment, for V2 in the SD3 treatment, and for V3 in the SD1 treatment. 

Grain filling and 1000-grain weight for the three cultivars in SD1 were higher than those in SD2, SD3, SD4, and SD5 for both years. The grain yield in SD1 was significantly higher than that in SD2, SD3, SD4, and SD5. SD5 had the lowest grain yield for all seeding dates. The grain yield for SD1 was 0.05–38.39% higher for V1, 3.34–29.01% higher for V2, and 0.07–41.23% higher for V3 than for SD2, SD3, SD4, and SD5, regarding the three cultivars. The panicles for V1 were 13.91% and 14.61% higher for V2 and V3, respectively. The spikelets per panicle were 12.12% and 0.55% higher for V2 than for V1 and V3, respectively. Grain filling was 10.37% and 1.42% higher for V3 than for V1 and V2, respectively. The 1000–grain weight was 12.21% and 8.05% higher for V3 than for V1 and V2, respectively. The grain yield was 19.16% and 6.67% higher for V3 than for V1 and V2, respectively. Significant differences in HI were also observed across sowing dates for all years and cultivars. HI was significantly higher for SD1 compared to SD2, SD3, SD4, and SD5 during both years and for all cultivars.

Although the rice yield for SD1 was higher than that for other treatments, the difference in yield between SD1 and SD2 did not reach significance. SD2 was sown seven days later than SD1, which overcame the tight connection between the previous and subsequent stubbles. Therefore, the three tested growth-constrained direct-seeded rice varieties may guarantee stable and large yields in late May, which is crucial for rice production in the hilly areas of Sichuan.

### 2.7. Principal Component Analysis of Growth Parameters under Different Sowing Dates

The main component was pre-and post–flowering temperature resources under SD1 and SD2 with abundant temperature resources (Figure 4). With a delay in the sowing date (SD3 and SD4), the temperature resources were gradually strained, and the main component was temperature resources at each stage. In SD5, it was greatly affected by low temperature; the main component was temperature resources at the earlier stage and panicles, and the temperature was the main influencing factor at different sowing dates.

### 2.8. Correlation of Rice Growth Parameters under Different Sowing Dates

The yield was positively correlated with spikelets per panicle, grain filling, 1000-grain weight, heading to maturity EAT, heading to maturity ADT, whole growth period ADT, full heading stage LAI, and maturity stage above-ground dry matter (Figure 5). The yield was negatively correlated with panicles and sowing-to-heading ADT. Whole growth period EAT was positively correlated with sowing to heading period days, heading to maturity days, and whole growth period days. The whole growth period ADT was negatively correlated with sowing to heading period days, heading to maturity days, and whole growth period days.

## 3. Materials and Methods

### 3.1. Rice Cultivars

Three medium indica rice cultivars, Luliangyoujingling (V1), Shenyouyuehesimiao (V2), and Huangyouyuehesimiao (V3), which are primarily cultivated for general rice production in Sichuan Province, were used in the field experiments. V1 was originally developed by the Rice and Sorghum Research Institute of the Sichuan Academy of Agricultural Sciences (RISAAS) and is an inbred indica rice with an average growth period of 142.6 days. V2 and V3 were developed by the Taiwo Agricultural Technology Co., Ltd. (Mianyang, China) (TATCL) and are hybrid indica rice cultivars with average growth periods of 127.9 and 127.2 days, respectively.

### 3.2. Experimental Treatments and Cultivation Methods

This research was performed in Dayan Town, Jiangyou City, Sichuan Province, China, between May and October of 2021 and 2022 (37°71′ N, 104°80′ E; annual average temperature: 16.2 °C; annual average sunshine: 1367 h; annual average rainfall: 1100 mm). The experimental field was a winter fallow cropland. The surface soil (0–20 cm depth) at the study site contained 1.87 g kg^−1^ total N [18] (Kjeldahl method, UDK-169, ITA), 26.98 mg kg^−1^ available phosphorus (Mo-Sb colorimetry following H_2_SO_4_ and HClO_4_ digestion), 2.51% organic matter (K_2_Cr_2_O_7_-volumetric method), and 58.16 mg kg^−1^ available K (flame spectrometry following NH_4_OAc extraction). The soil pH was 6.69, as measured in a sample with a 1:2.5 soil:water ratio. Average air temperature and precipitation data from the previous rice-growing season (Figure 1) were collected from a weather station close to the experimental site.

The experimental design used for the study was a two-factor split-plot design with three replicates per treatment. The experimental sowing period was the main plot, with cultivars as the treatment for the subplots (length and width: 5 × 6 m). The main plot sowing periods were SD1 (16 May), SD2 (23 May), SD3 (30 May), SD4 (6 June), and SD5 (13 June). The wet direct-seeding rice method was implemented. Seeds of each variety were planted at a density of 23.5 kg ha^−1^.

Germinated seeds were selected for seeding, with approximately three to four seeds per hole. The row spacing was 0.36 × 0.18 m and uniform seedlings at the three-four-leaf stage were planted, with one plant per hole. A plot size of 30 m^2^ was used in both years. Because adjacent plots had different sowing dates, they were isolated by a main interval ridge, which reduced fertilizer and waterside seepage. The height and base of the plot ridge were set at 0.30 m, and an independent single-row drainage and irrigation system was established for fertilizer water management. 

Fertilizer management was the same in both years. Nitrogen, phosphorus, and potassium fertilizers were applied in the same way for each treatment. The amount of specific compound fertilizer for rice (N-P_2_O_5_-K_2_O = 26-9-9) was converted to be equal to 180 kg of pure N per ha and then used uniformly. Three different N management models were established in which base, tiller, and panicle fertilizers were applied at a 5:3:2 ratio. Water management was based on rice alternate dry-wet irrigation. Water was drained one week before harvest. In all plots, weeds were controlled with pre-emergence herbicides and hand weeding. Pesticides and fungicides were sprayed to control pests and diseases if needed. Other methods of cultivation management were performed according to local high-yield cultivation methods.

### 3.3. Investigation of the Growth Period and Determination of Effective Accumulated Temperature

The sowing, heading, and maturity dates were recorded accurately, and the sowing-heading date, heading-maturing date, and total growth period were calculated. The effective accumulated temperature for each growth stage was calculated as follows: $Ke=\sum_{i}^{n}T1-T0$, where T1 is the average daily temperature on day i and T0 is the developmental threshold temperature of the plant, which is 10 °C [19,20].

### 3.4. Yield and Yield Components

At maturity, rice grain was collected from a 5 m^2^ sampling area in each subplot and used to calculate grain yield, which was then adjusted to a 13.5% safe moisture content by weight. Grain yield components were measured, including the 1000-grain weight was calculated from three replicates; grain filling and panicles (containing the number of full grains and the number of empty grains) were measured.

### 3.5. Total Dry Weight (TDW)

In both years, five hills of plants were sampled from each subplot to determine the dry weight at the heading and maturity stages, and then all plants were divided into the leaf, stem sheath, and panicle. The plants were treated at 105 °C for 30 min, then the dry weight of each part of the plant was determined after oven drying at 80 °C for 72 h to achieve a constant weight. The dry weights of all plant parts were calculated relative to the biomass of each growing stage.

### 3.6. Determination of Leaf Area Index and Chlorophyll Content of Rice

#### 3.6.1. Determination of Chlorophyll Content in Rice

Five flag leaves from rice with the same growth were selected at the beginning of the full heading, then again 7, 14, and 21 days after the full heading. The five flag leaves were extracted with acetone and absolute ethanol.

#### 3.6.2. Determination of Leaf Area Index of Rice

The total green leaf area per plant was measured with an LI-3100C leaf area analyzer (American LI-COR, Lincoln, NE, USA) during the tillering and full heading stages in both years. The leaf area index is the multiple of the total leaf area of plants in the land area per unit land area.

### 3.7. Statistical Analysis

The data were analyzed using analysis of variance (ANOVA), and means were compared based on the least significant difference (LSD) test at the 0.05 probability level using SPSS 23 (Statistical Product and Service Solutions Inc., Chicago, IL, USA). Figures were prepared using Origin Pro 2017 (OriginLab, Northampton, MA, USA).

## 4. Discussion

### 4.1. Effects of the Sowing Date on the Growth and Temperature of Growth-Constrained Direct-Seeding Rice

Direct-seeding rice avoids the necessity for raising seedlings in multi-cropping areas, and rice is often sown after the previous crop has been harvested. The essential stages of rice growth (sowing, jointing, heading, and maturity) are shortened to varying degrees, the total growth period is shortened, and the practice also results in reduced utilization of climate resources, such as temperature and light [21,22]. The results of this study are consistent with those of previous studies. The total growth period of direct-seeding rice was shortened by delaying the sowing date (Table 1). During the appropriate growing season, the growth process is primarily controlled by the rice itself [23,24]. Different rice varieties responded to changes in the sowing date, and the length and distribution of each growth stage are different. Studies have reported the effect of sowing date on rice growth characteristics; however, the results have been inconsistent or limited because of differences in the planting area, growing season, variety, and research methods [14,25]. The early growth process was sometimes accelerated as the sowing date was delayed; however, some studies showed that the total growth period remained the same, even when the sowing date was delayed [26]. In the present study, the total growth period of indica rice was slightly shortened when the sowing date was delayed. Rice growth is closely related to the distribution of temperature and light resources, with temperature being the primary environmental factor that affects growth [27,28]. In recent years, EAT has been used in crop growth [29]. Effective accumulated temperature is the main index affecting rice growth [30]. It could be improved the yield of rice in effective cumulative temperature and a longer growth duration rice season [31,32]. In this study, when the sowing date was delayed, the effective accumulated temperature during the vegetative growth period of indica rice changed irregularly; however, during the total growth period, the effective accumulated temperature of direct-seeding rice decreased. Panicles are initiated when the effective accumulated temperature reaches a specific value, indicating that indica rice is a temperature-sensitive rice (Figure 4). The safe temperature index for full heading is an average daily temperature ≤20 °C (japonica) or 22 °C (indica) over three days, and full heading will not occur below this temperature. According to this temperature standard, the safe heading date for indica rice in the Sichuan Hill area was before 28 August 2021 and before 7 September 2022 (Figure 1). To ensure the safe full heading of indica rice, sowing should occur in late May (SD2), which is consistent with local practices. Lu et al. [33] performed a limited sowing date experiment to explore the potential of delaying the sowing date of indica–japonica hybrid rice (Yongyou 1540). They found that delaying sowing to August 16 still resulted in a full panicle (on 6 October) and a mature harvest. To ensure a 9 t ha^−1^ yield of rice, the theoretical limit is before 12 July. In this study, to ensure a 9 t ha^−1^ yield of direct-seeding hybrid rice, the limit was before 30 May. This was due to the difference between the experimental and the theoretical limits of the sowing date.

### 4.2. Effects of Different Seeding Dates on the Growth and Grain Yield of Growth-Constrained Direct-Seeding Rice

A delayed sowing date affected seedling growth in the early stages and vegetative growth at a later stage [34,35,36]. By delaying the sowing date, the dry matter mass decreased, especially the accretion of dry matter during the early stage, and resulted in a decreased yield [37,38]. The results showed that delaying the sowing date shortened the vegetative growth period and the population of rice. The shortening of the booting stage decreased the number of spikelets and grains, and delaying the heading date decreased the daily average temperature and effective accumulated temperature at the filling stage, which led to lower grain filling [39,40]. In this study, it was greatly affected by low temperatures; the main component was temperature resources during the earlier stage and panicles, and the temperature was the main influencing factor at different sowing dates [41]. Under sowing SD1 treatment, indica–japonica hybrid rice had a higher dry matter quality accumulation and a higher harvest index at the vegetative growth stage, resulting in a yield higher than obtained with other sowing dates (Table 3 and Table 4). The change in the sowing date affects the formation of yield by affecting the synthesis of photosynthetic substances in rice [42]. The yield was positively correlated with the maturity stage of above-ground dry matter (Figure 5). The low tolerance of indica rice to low temperatures decreased the harvest index as the sowing date was delayed, and it became difficult to complete the panicle safely. This resulted in a significant reduction in the seed setting rate and increased the difference in the yield of hybrid rice. The yield of rice decreased because of the increase in daily average temperature in the early stage, resulting in the shortened vegetative growth period, and the temperature was decreased after the heading stage, which led to the decrease of sink capacity and grain filling of hybrid rice [43].

## 5. Conclusions

In the hilly rice-producing region of Sichuan, indica hybrid rice adapts well to different sowing dates, and the difference in yield between SD1 and SD2 treatments was not significant. This was due to increased material accumulation after flowering and increased assimilative material transport capacity compared with other treatments. Indica hybrid rice is highly photosensitive and tolerant of low temperatures. Among the three varieties tested, V3 had the highest yield with SD2. When the previous crop harvest was late, and the sowing date was SD2, the indica–japonica hybrid rice had the highest guaranteed yield. This two-year study also found that the indica–japonica hybrid rice yield can be maintained at a high level with a sowing date between 16 May and 23 May each year.

## Figures and Tables

**Figure 1 plants-12-01899-f001:**
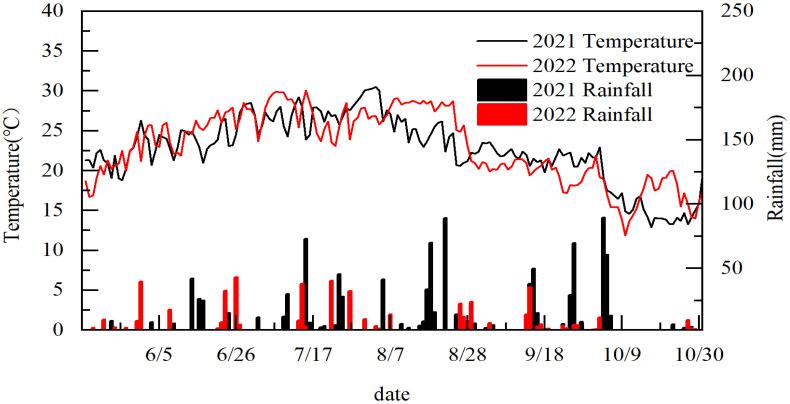
Meteorological data of rainfall and temperature during the experiment.

**Figure 2 plants-12-01899-f002:**
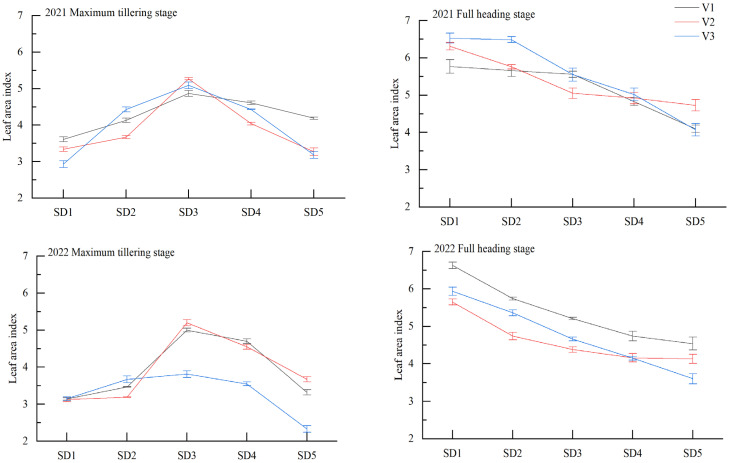
Effect of sowing date on leaf area index of hybrid rice in main growth stages. V1, luliangyoujingling; V2, shenyouyuehesimiao; V3, huangyouyuehesimiao; SD1, 16 May; SD2, 23 May; SD3, 30 May; SD4, 6 June; SD5, 13 June.

**Figure 3 plants-12-01899-f003:**
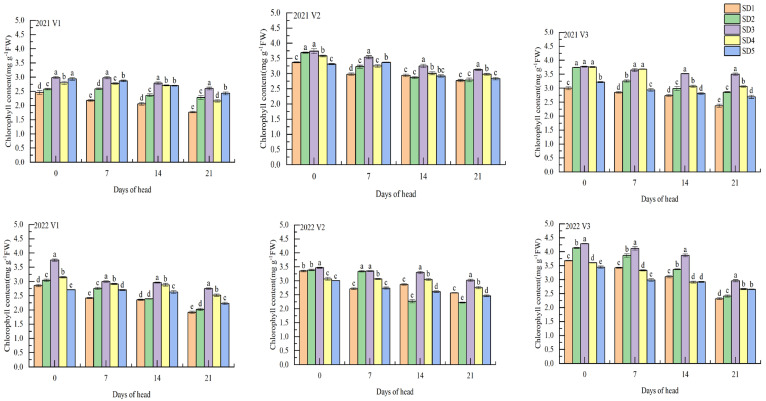
Effects of the sowing date on chlorophyll content and peroxidase activity in flag leaves of growth-constrained direct-seeded rice after full heading. V1, luliangyoujingling; V2, shenyouyuehesimiao; V3, huangyouyuehesimiao; SD1, 16 May; SD2, 23 May; SD3, 30 May; SD4, 6 June; SD5, 13 June. Different letters are significantly different according to LSD (0.05).

**Figure 4 plants-12-01899-f004:**
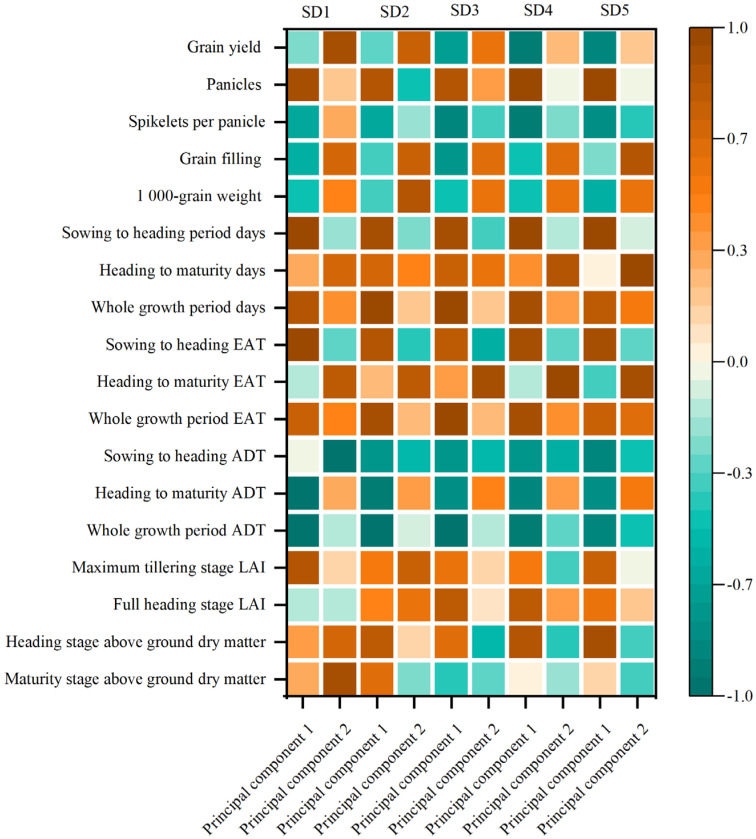
Effects of sowing date on principal component load matrix of rice from 2021 to 2022. EAT, effective accumulated temperature; ADT, average daily temperature; LAI, Leaf area index, SD1, 16 May; SD2, 23 May; SD3, 30 May; SD4, 6 June; SD5, 13 June.

**Figure 5 plants-12-01899-f005:**
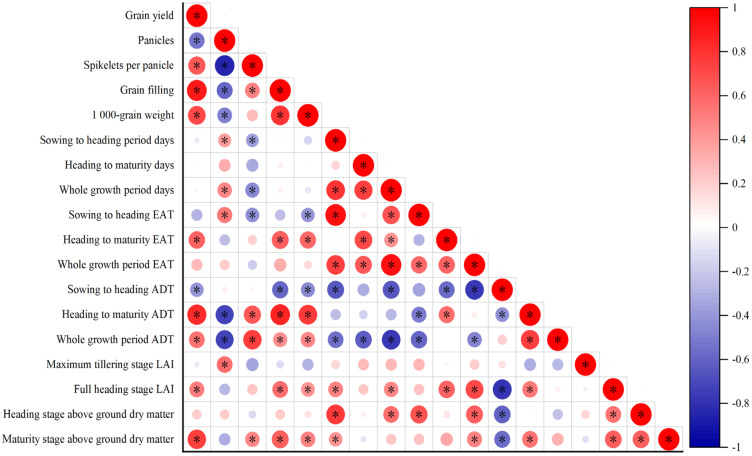
Correlation of rice growth parameters under different sowing dates. EAT, effective accumulated temperature; ADT, average daily temperature; LAI, Leaf area index,* significances at *p* < 0.05, respectively.

**Table 1 plants-12-01899-t001:** Effect of sowing date on the growth process of direct seeding rice.

Year	Cultivar	Sowing Date	Sowing to Heading Period/Days	Heading to Maturity Period/Days	Whole Growth Period /Days
2021	V1	SD1	103 a	48 b	151 a
		SD2	100 ab	50 ab	150 a
		SD3	97 bc	52 a	149 a
		SD4	96 bc	50 ab	146 a
		SD5	95 c	44 c	139 b
		Mean	98	49	147
	V2	SD1	89 a	61 b	150 a
		SD2	88 ab	61 b	149 a
		SD3	85 abc	64 a	149 a
		SD4	84 bc	62 ab	146 a
		SD5	82 c	57 c	139 b
		Mean	86	61	147
	V3	SD1	82 a	55 a	137 a
		SD2	81 a	55 a	136 a
		SD3	81 a	55 a	136 a
		SD4	78 ab	54 ab	132 ab
		SD5	77 b	52 b	129 b
		Mean	80	54	134
2022	V1	SD1	92 a	39 b	131 a
		SD2	87 b	44 a	131 a
		SD3	86 bc	44 a	130 a
		SD4	83 c	43 a	126 ab
		SD5	82 c	39 b	121 b
		Mean	86	42	128
	V2	SD1	87 a	38 c	125 a
		SD2	81 b	39 c	120 b
		SD3	78 b	42 b	120 b
		SD4	72 c	48 a	120 b
		SD5	72 c	47 a	119 b
		Mean	78	43	121
	V3	SD1	79 a	41 c	120 a
		SD2	78 a	38 b	116 ab
		SD3	77 a	39 b	116 ab
		SD4	71 b	43 a	114 b
		SD5	71 b	43 a	114 b
		Mean	75	41	116
F value		Y	11,438 **	44,415 **	31,374 **
		C	1855 **	569 **	748 **
		SD	3398 **	1014 **	1203 **
		Y × C	89 **	646 **	77 **
		Y × SD	175 **	590 **	90 **
		C × SD	22 **	126 **	29 **
		Y × C × SD	17 **	51 **	9 **

V1, luliangyoujingling; V2, shenyouyuehesimiao; V3, huangyouyuehesimiao; SD1, 16 May; SD2, 23 May; SD3, 30 May; SD4, 6 June; SD5, 13 June. Y, year; C, cultivar; SD, sowing date; means followed by different letters are significantly different according to LSD (0.05) and ** significances at *p* < 0.01, respectively.

**Table 2 plants-12-01899-t002:** Effects of sowing date on effective accumulated temperature and the daily average temperature of hybrid rice.

Year	Cultivar	Sowing Date	Sowing to Heading		Heading to Maturity		Whole Growth Period	
			EAT/°C	ADT/°C	EAT/°C	ADT/°C	EAT/°C	ADT/°C
2021	V1	SD1	1576.4 a	25.3 bc	562.2 a	21.7 a	2138.6 a	24.2 a
		SD2	1547.3 a	25.5 b	540.0 ab	20.8 b	2087.3 a	23.9 ab
		SD3	1530.0 a	25.8 ab	523.3 b	20.1 c	2053.3 ab	23.8 b
		SD4	1544.8 a	26.1 a	431.7 c	18.6 d	1976.5 b	23.5 bc
		SD5	1519.4 a	26.0 a	358.0 d	18.1 e	1877.4 c	23.5 c
		Mean	1543.6	25.7	483	19.9	2026.6	23.8
	V2	SD1	1373.4 a	25.4 b	773.5 a	22.7 a	2146.9 a	24.3 a
		SD2	1382.0 a	25.7 b	700.4 b	21.5 b	2082.4 a	24.0 ab
		SD3	1371.0 a	26.1 a	683.3 c	20.7 c	2054.3 ab	23.8 b
		SD4	1350.3 a	26.1 a	625.5 d	20.1 d	1975.8 b	23.5 bc
		SD5	1323.9 a	26.1 a	557.4 e	19.8 e	1881.3 c	23.5 bc
		Mean	1360.1	25.9	668	21.0	2028.1	23.8
	V3	SD1	1255.1 a	25.3 c	785.0 a	24.3 a	2040.1 a	24.9 a
		SD2	1277.1 a	25.8 b	728.4 b	23.2 b	2005.5 a	24.7 ab
		SD3	1307.8 a	26.1 a	668.6 c	22.2 c	1976.4 ab	24.5 b
		SD4	1274.5 a	26.3 a	625.4 d	21.6 d	1899.9 b	24.4 b
		SD5	1255.1 a	26.3 a	566.3 e	20.9 e	1821.4 c	24.1 c
		Mean	1273.9	26.0	674.7	22.4	1948.7	24.5
2022	V1	SD1	1447.3 a	25.7 c	484.0 b	22.4 a	1931.3 a	24.7 a
		SD2	1421.6 ab	26.3 b	510.4 a	21.6 b	1932.0 a	24.7 a
		SD3	1450.9 a	26.9 a	449.3 c	20.2 c	1900.2 a	24.6 a
		SD4	1411.6 ab	27.0 a	401.6 d	19.3 d	1813.2 b	24.4 ab
		SD5	1379.5 b	26.8 a	345.3 e	18.9 e	1724.8 c	24.3 b
		Mean	1422.2	26.5	438.1	20.5	1860.3	24.5
	V2	SD1	1354.2 a	25.6 d	521.3 a	23.7 a	1875.5 a	25.0 ab
		SD2	1309.9 a	26.2 c	524.4 a	23.4 a	1834.3 a	25.3 a
		SD3	1304.9 a	26.7 b	508.5 a	22.1 b	1813.4 a	25.1 ab
		SD4	1225.2 b	27.0 b	499.1 a	21.6 c	1724.3 b	24.9 b
		SD5	1256.4 b	27.5 a	460.4 b	19.8 d	1716.8 b	24.4 c
		Mean	1290.1	26.6	514.7	22.1	1804.9	24.9
	V3	SD1	1212.1 ab	25.3 d	612.0 a	24.9 a	1824.1 a	25.2 a
		SD2	1254.6 a	26.1 c	536.3 b	24.1 b	1790.9 a	25.4 a
		SD3	1286.1 a	26.7 b	495.6 c	22.7 c	1781.7 ab	25.4 a
		SD4	1206.8 b	27.0 b	517.1 b	22.0 d	1723.9 b	25.1 a
		SD5	1237.7 ab	27.4 a	457.2 d	20.6 e	1694.9 b	24.9 b
		Mean	1239.5	26.5	523.6	22.9	1763.1	25.2
F value		Y	5227.41 **	3543.27 **	53,423.54 **	3325.32 **	18,941.42 **	6012.69 **
		C	422.11 **	1085.59 **	12,302.09 **	10,587.05 **	3012.80 **	375.72 **
		SD	4496.93 **	7.14 **	7841.31 **	4234.85 **	457.24 **	431.46 **
		Y × C	20.13 **	146.50 **	1063.86 **	345.58 **	126.30 **	38.90 **
		Y × SD	118.56 **	15.78 **	1056.33 **	96.43 **	34.78 **	32.97 **
		C × SD	38.62 **	9.11 **	181.37 **	45.84 **	17.56 **	7.25 **
		Y × C × SD	28.58 **	5.49 **	103.32 **	40.40 **	6.62 **	4.19 **

V1, luliangyoujingling; V2, shenyouyuehesimiao; V3, huangyouyuehesimiao; EAT, effective accumulated temperature; ADT, average daily temperature. SD1, 16 May; SD2, 23 May; SD3, 30 May; SD4, 6 June; SD5, 13 June. Y, year; C, cultivar; SD, sowing date; means followed by different letters are significantly different according to LSD (0.05)and ** significances at *p* < 0.01, respectively.

**Table 3 plants-12-01899-t003:** Dry matter accumulation of the above-ground part of hybrid rice in the main growth period(t ha^−1^).

Year	Cultivar	Sowing Date	Heading Stage		Maturity Stage		Heading to Maturity	Dry Matter Increase of Panicle
			Stem and Leaf	Panicle	Stem and Leaf	Panicle		
2021	V1	SD1	11.72 a	2.38 a	7.68 a	10.61 a	4.20 a	8.24 a
		SD2	10.75 b	2.27 b	7.61 a	9.12 b	3.71 b	6.85 b
		SD3	10.46 b	2.13 c	7.55 a	8.15 c	3.11 c	6.01 d
		SD4	10.31 b	2.00 d	7.31 b	8.01 c	3.01 c	6.02 d
		SD5	8.85 c	1.54 e	6.28 c	7.88 c	3.76 b	6.33 c
		Mean	10.42	2.06	7.29	8.75	3.56	6.69
	V2	SD1	7.95 a	1.77 a	6.54 a	10.25 a	7.07 a	8.48 a
		SD2	7.93 a	1.72 ab	6.10 b	9.86 a	6.31 b	8.14 b
		SD3	7.74 ab	1.66 b	6.01 b	8.26 b	4.86 c	6.59 c
		SD4	7.55 ab	1.56 c	5.99 b	7.86 c	4.75 c	6.30 c
		SD5	7.47 b	1.50 c	4.72 c	7.36 d	3.11 d	5.86 d
		Mean	7.73	1.64	5.87	8.72	5.22	7.08
	V3	SD1	9.81 a	2.30 a	6.92 a	10.72 a	5.53 a	8.42 a
		SD2	9.48 a	2.24 a	5.62 b	9.90 b	3.80 bc	7.67 b
		SD3	7.39 b	1.76 b	5.27 c	9.06 c	5.18 a	7.30 c
		SD4	7.19 b	1.69 b	4.84 d	8.04 d	4.01 b	6.36 d
		SD5	6.53 c	1.55 c	4.49 e	7.07 e	3.48 c	5.52 e
		Mean	8.08	1.91	5.43	8.96	4.40	7.05
2022	V1	SD1	7.36 c	1.35 b	5.59 d	9.36 ab	6.25 a	8.02 a
		SD2	9.21 a	1.74 a	6.74 a	9.72 a	5.51 b	7.98 a
		SD3	9.01 a	1.41 b	6.36 b	9.04 b	4.99 c	7.64 b
		SD4	9.00 a	1.36 b	6.04 c	7.16 c	2.84 d	5.80 c
		SD5	8.71 b	1.34 b	5.35 d	6.71 d	2.01 e	5.37 d
		Mean	8.66	1.44	6.02	8.40	4.32	6.96
	V2	SD1	8.75 a	1.53 a	6.02 a	9.58 a	5.33 ab	8.06 a
		SD2	7.82 b	1.43 b	5.56 b	9.39 a	5.71 a	7.96 a
		SD3	7.68 b	1.35 c	5.54 b	8.50 b	5.01 b	7.15 b
		SD4	7.23 c	1.27 d	5.06 c	8.47 b	5.04 b	7.20 b
		SD5	6.84 d	1.12 e	4.54 d	7.70 c	4.29 c	6.59 c
		Mean	7.66	1.34	5.35	8.73	5.08	7.39
	V3	SD1	8.94 a	2.15 a	5.81 a	11.26 a	5.98 d	9.11 b
		SD2	7.95 b	1.64 b	6.00 a	10.04 b	6.45 c	8.40 c
		SD3	7.55 b	2.10 a	6.10 a	11.02 a	8.47 a	9.93 a
		SD4	6.59 c	1.27 c	5.92 a	9.49 c	7.54 b	8.22 c
		SD5	6.55 c	1.15 d	5.02 b	8.68 d	6.00 d	7.52 d
		Mean	7.52	1.66	5.77	10.10	6.89	8.64
F value		Y	20.04 **	3147.70 **	62.02 **	111.57 **	280.35 **	737.11 **
		C	2476.34 **	1690.77 **	2241.73 **	5613.34 **	1132.36 **	3880.69 **
		SD	1594.06 **	607.39 **	1091.70 **	736.28 **	1151.18 **	865.59 **
		Y × C	82.86 **	96.89 **	87.34 **	231.95 **	61.34 **	153.98 **
		Y × SD	573.74 **	228.71 **	422.39 **	199.34 **	392.02 **	170.00 **
		C × SD	162.13 **	49.14 **	149.87 **	143.60 **	69.60 **	128.60 **
		Y × C × SD	114.39 **	110.17 **	48.56 **	102.51 **	81.96 **	85.21 **

V1, luliangyoujingling; V2, shenyouyuehesimiao; V3, huangyouyuehesimiao; SD1, 16 May; SD2, 23 May; SD3, 30 May; SD4, 6 June; SD5, 13 June. Y, year; C, cultivar; SD, sowing date; means followed by different letters are significantly different according to LSD (0.05) and ** significances at *p* < 0.01, respectively. Heading to maturity, heading to maturity is related to dry matter accumulation from the full heading stage to the mature stage.

**Table 4 plants-12-01899-t004:** Effects of sowing date on yield and yield components of Hybrid Rice.

Year	Cultivar	Sowing Date	Panicles (104 ha^−1^)	Spikelets per Panicle	Grain Filling (%)	1000-Grain Weight (g)	Grain Yield (t ha^−1^)	HI
2021	V1	SD1	278.31 d	161.97 a	87.48 a	26.04 a	10.27 a	0.57 a
		SD2	287.21 c	158.68 a	85.47 a	25.40 a	9.90 a	0.58 a
		SD3	354.76 a	124.19 b	74.50 b	24.80 b	8.14 b	0.57 ab
		SD4	338.71 b	122.39 b	72.81 b	24.11 c	7.28 c	0.53 bc
		SD5	323.21 b	121.78 b	72.41 b	23.92 c	6.82 d	0.46 c
		Mean	316.44	137.80	78.53	24.85	8.48	0.53
	V2	SD1	235.78 c	199.66 a	86.29 a	25.70 a	10.44 a	0.61 a
		SD2	244.01 c	194.60 a	83.97 a	25.03 ab	9.98 ab	0.60 a
		SD3	315.98 a	154.52 b	81.98 b	24.24 b	9.70 b	0.60 a
		SD4	303.22 ab	144.00 c	80.34 bc	24.20 b	8.49 c	0.53 b
		SD5	289.77 b	137.10 d	77.54 c	24.16 b	7.44 d	0.53 b
		Mean	277.75	165.98	82.02	24.67	9.21	0.57
	V3	SD1	221.96 c	178.33 a	90.14 a	29.47 a	10.52 a	0.69 a
		SD2	230.53 c	175.97 a	88.97 a	28.25 b	10.19 a	0.65 b
		SD3	336.09 a	136.20 b	83.33 b	26.25 c	10.01 a	0.63 b
		SD4	297.97 b	134.41 b	76.57 c	26.12 c	8.01 b	0.63 b
		SD5	286.26 b	127.79 c	76.35 c	25.49 c	7.12 c	0.61 c
		Mean	274.56	150.54	83.07	27.12	9.17	0.64
2022	V1	SD1	226.09 d	170.85 c	82.96 a	25.87 a	8.29 a	0.55 b
		SD2	261.48 c	194.87 a	74.42 b	22.56 b	8.56 a	0.58 a
		SD3	274.16 b	186.22 b	73.05 b	22.55 b	8.41 a	0.52 c
		SD4	303.54 a	156.21 d	72.51 b	22.52 b	7.57 b	0.49 d
		SD5	303.38 a	154.69 d	62.44 c	22.20 b	6.59 c	0.47 d
		Mean	273.73	172.57	73.08	23.14	7.88	0.52
	V2	SD1	231.48 b	177.21 a	88.35 a	26.80 a	9.61 a	0.60 b
		SD2	237.65 ab	182.23 a	86.07 a	25.28 b	9.42 a	0.64 a
		SD3	239.99 ab	184.12 a	84.82 ab	24.66 b	9.24 a	0.64 a
		SD4	247.59 a	183.58 a	81.48 b	24.63 b	8.99 b	0.63 a
		SD5	245.14 a	182.90 a	74.10 c	24.47 b	8.10 c	0.56 c
		Mean	240.37	182.01	82.96	25.17	9.07	0.61
	V3	SD1	216.30 c	215.14 a	91.91 a	26.55 b	11.13 a	0.66 a
		SD2	222.22 c	198.89 b	91.73 a	28.31 a	11.31 a	0.64 a
		SD3	261.56 a	195.73 b	88.16 a	27.43 ab	11.38 a	0.62 b
		SD4	263.95 a	185.26 c	75.69 b	25.77 bc	9.54 b	0.61 bc
		SD5	237.86 b	182.80 c	73.80 b	25.60 c	8.21 c	0.59 c
		Mean	240.38	195.56	84.26	26.73	10.31	0.62
F value		Y	5141.01 **	12,723.47 **	471.54 **	36.82 **	70.96 **	67.80 **
		C	1048.51 **	1504.78 **	2242.48 **	490.98 **	2586.21 **	397.60 **
		SD	3773.84 **	886.14 **	826.10 **	1723.22 **	508.27 **	345.69 **
		Y × C	552.66 **	1046.53 **	372.08 **	17.09 **	107.16 **	28.98 **
		Y × SD	77.91 **	328.86 **	101.39 **	42.96 **	94.11 **	48.50 **
		C × SD	145.08 **	80.99 **	50.03 **	13.41 **	59.41 **	16.00 **
		Y × C × SD	36.24 **	87.55 **	55.91 **	19.13 **	40.95 **	17.95 **

V1, luliangyoujingling; V2, shenyouyuehesimiao; V3, huangyouyuehesimiao; EAT, effective accumulated temperature; ADT, average daily temperature; SD1, 16 May; SD2, 23 May; SD3, 30 May; SD4, 6 June; SD5, 13 June. Y, year; C, cultivar; SD, sowing date; means followed by different letters are significantly different according to LSD (0.05) and ** significances *p* < 0.01, respectively.

## Data Availability

Not applicable.

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
