# Peer review of "Effect of Sowing Date on the Growth Characteristics and Yield of Growth-Constrained Direct-Seeding Rice"

_plants, 2023, doi:10.3390/plants12091899_

Round 1

Reviewer 1 Report (Previous Reviewer 1)

Comments to authors

Manuscript ID: plants-2331202

This is the second time that I have evaluated this manuscript. Unfortunately, there are still many questions about the treatments of this study that prevent its acceptance for publication, which are as follows:

1-      The treatments of sowing dates are not logical. There are only 7 days intervals between each two sowing dates. The five treatments of sowing date can be reduced to only three treatments and can be mentioned as early sowing date, normal sowing date, and late sowing date.

2-      Based on the information mentioned in the M&M section, V1 and V2 represent late maturity (142.6 days) and early maturity (127.9 days) genotypes, respectively. However, in the result section, the maturity dates of the two cultivars were similar in the first year and very close in the second year. Why? 

3-      Although the differences in the daily average temperature between two years for V1, V2, and V3 were only 0.7, 1.1, and 0.7 °C (Table 2), the differences in the whole growth period (days) between the first and second years were 19, 26, and 18 days (Table 1), respectively. This means that the temperature is not the main reason for these differences.

4-      Lines 159-161: How the authors determined the total leaf area of plants although they measured leaf area for only three leaves.    

5-       The manuscript needs a huge language editing in many places

Minor comments (there are many minor comments in the manuscript) but this only a few examples).

1-      Figure 2: change infex into index

2-      Lines 19-20: the physiological characteristics of yield, There is no term in crop science with this meaning

3-      Line 26: Remove the dot between a ton and a hectare

4-      Title of Table 3 “Dry matter accumulation matter accumulation” the word “matter accumulation” has been repeated.  

5-      Peroxidase activity is not mentioned in the title of Figure 3.

6-      What is the meaning of “highest tillering stage” the authors mean the maximum tillering stage.

Author Response

Point 1: The treatments of sowing dates are not logical. There are only 7 days intervals between each two sowing dates. The five treatments of sowing date can be reduced to only three treatments and can be mentioned as early sowing date, normal sowing date, and late sowing date.

Response 1: Thank you for your comment.There are 7 days intervals between each two sowing dates is better to observe the regularity and temperature resource changes.May 16 (SD1) is normal sowing date.

Point 2: Based on the information mentioned in the M&M section, V1 and V2 represent late maturity (142.6 days) and early maturity (127.9 days) genotypes, respectively. However, in the result section, the maturity dates of the two cultivars were similar in the first year and very close in the second year. Why? 

Response 2: Thank you for your comment.The temperature in the second year is higher than that in the first year during rice season.

Point 3: Although the differences in the daily average temperature between two years for V1, V2, and V3 were only 0.7, 1.1, and 0.7 °C (Table 2), the differences in the whole growth period (days) between the first and second years were 19, 26, and 18 days (Table 1), respectively. This means that the temperature is not the main reason for these differences.

Response 3: Thank you for your comment.Temperature is one of the factors affecting sowing date.

Point 4:Lines 159-161: How the authors determined the total leaf area of plants although they measured leaf area for only three leaves. 

Response 4: Thank you for your comment.We have already revised it,three leaves modify total green leaf area per plant(line 159)

Point 5:The manuscript needs a huge language editing in many places

Minor comments (there are many minor comments in the manuscript) but this only a few examples).

1-Figure 2: change infex into index

2-Lines 19-20: the physiological characteristics of yield, There is no term in crop science with this meaning

3-Line 26: Remove the dot between a ton and a hectare

4-Title of Table 3 “Dry matter accumulation matter accumulation” the word “matter accumulation” has been repeated.  

5-Peroxidase activity is not mentioned in the title of Figure 3.

6-What is the meaning of “highest tillering stage” the authors mean the maximum tillering stage.

 Response 5:  Thank you for your comment.We have already revised it,infex modify index,delete physiological,delete matter accumulation,highest modify maximum.(line 20,245-264,321,351)

Reviewer 2 Report (Previous Reviewer 2)

The manuscript entitled: Effect of sowing date on the growth characteristics and yield of growth-constrained direct-seeding rice has been improved. I included some comments in the original text (pdf). After corrections, I recommend publishing in the journal Plants.

Author Response

Reviewer 3 Report (Previous Reviewer 3)

Hello, 

thank you for accepting my previous comments. Still the author's team to big for the work presented here. 

Round 2

Reviewer 1 Report (Previous Reviewer 1)

1-    The manuscript needs a huge language editing in many places

2-    Table 2: the abbreviations are not explained correctly in the footnote of Table 2. Please check EAT and ADT in the footnote of the Table. The same problem is found in Figures 4 and 5.

3-    Figure 2: it is better to convert the columns into lines to show the highest and lowest points for LAI with different sowing dates.

4-    Line 491: Lu[28] et al. please write the reference in the correct form.

5-    Line 494-495: 9 t· ha-1, remove dots between t and ha. In all the manuscript. Please follow the guideline of the journal for units 

Author Response

Point 1: The manuscript needs a huge language editing in many places

Response 1:  Thank you for your comment.We have already revised it.This document certifies that the manuscript listed below has been edited to ensure language and grammar accuracy and is error free in these aspects.

Point 2: Table 2: the abbreviations are not explained correctly in the footnote of Table 2. Please check EAT and ADT in the footnote of the Table. The same problem is found in Figures 4 and 5.

Response 2:  Thank you for your comment.We have already revised it.(line 229,230,391,392,432,458)

Point 3: Figure 2: it is better to convert the columns into lines to show the highest and lowest points for LAI with different sowing dates.

Response 3:  Thank you for your comment.We have already revised it.Because some values are close to each other, the variance analysis letters will overlap, or overlap with the error bar, so there is no variance analysis letter in the figure.(line 244-266)

Point 4: Line 491: Lu[28] et al. please write the reference in the correct form.

Response 4:  Thank you for your comment.We have already revised it.(line 493,494)

Point 5: Line 494-495: 9 t· ha-1, remove dots between t and ha. In all the manuscript. Please follow the guideline of the journal for units

Response 5:  Thank you for your comment.We have already revised it.(line 26,96-98,112,327,352,390,496,497)

This manuscript is a resubmission of an earlier submission. The following is a list of the peer review reports and author responses from that submission.

Round 1

Reviewer 1 Report

Comments to authors

Manuscript ID: plants-2309233  

The authors aimed to investigate the differences in the physiological, growth, dry matter accumulation, and yield characteristics of three popular indica rice varieties under five sowing dates. My comments about this manuscript are the following:

1.      Please use other keywords did not mention in the title.

2.      Line 38: added a comma after rice (rice, and green manure–rice).

3.      Line 38: please change planting modes to cropping rotation

4.      Lines 79-82: What is this? Care must be taken when preparing your manuscript. Please follow the Instructions for the Authors.

5.      Lines 105-107: The treatments of sowing dates are not logical. There are only 7 days intervals between each two sowing dates. At least 15 days are acceptable as intervals between each two sowing dates. Additionally, which of time from the five sowing dates is the best sowing time for rice in Dayan Town, Jiangyou City?

6.      Line 108: please use SI Units (International System of Units) in all manuscript

7.      There is no mention of Figure 1 in the text

8.      Lines 127-128: what means the T0 in the equation of effective accumulated temperature?

9.      Line 135: This reference is not written correctly (Wang.etal[17].  Please follow the Instructions for the Authors.

10.  Lines 132-135: : the methods for measurements of all traits mentioned in Lines 132-135 did not need a reference. Please delete reference mentioned in line 135.

11.  Line 109 “hill” and line 110 “hole”, what is the difference between the two words?  

12.  Lines 109-110: there are three to four seeds per hill while there is one seedling per hole. There is a conflict in information between both.

13.   Check mistake in the line 139 .

14.  Lines 145-147: There is no relation between the method for measuring chlorophyll content and reference No. 18.

15.  Line 148: the subtitle is “Determination of leaf area index of rice”, while the information in the text of this subtitle (Lines 149-151) is not related to calculating the LAI.

16.  In M&M section, the authors did not provide any information about irrigation during the growing season such as the amount of irrigation water applied and the irrigation system used to apply the irrigation.

17.  In table 1, there is a Chinese character next to F, Why?

18.  In Table 1, the differences in the whole growth period (days) between the first and second years for V1, V2, and V3 were 19, 26, and 18 days, respectively. Why this big differences between the two years? Additionally, the average growth period for V1 (LLYJL) and V2 (SYYHSM) were 142.6 and 127.9 days (as mentioned in M&M section, Lines 90-92). However, the whole growth period for V1 and V2 were 147 and 147 days in first year and 128 and 121 days in the second year, respectively (as mentioned in Table 1). The same question why these differences?

19.  In Table 1, Table 2: what is the meaning of Y, C, and SD? Please explain these abbreviations under each table.

20.  Lines 2018-2019: what is the meaning “highest tillering stage” please used the Zadoks scale to explain this stage correctly.

21.  There is a mistake in the title of Table 3 “matter accumulation Matter Accumulation”.

22.  Table 3, the values in the column “heading to maturity” is related to what?

23.  References are not formatted according to the journal’s criteria.

Reviewer 2 Report

Interesting research results for science and agricultural practice. I appreciate that this is a two-year field experience. However, the manuscript needs improvement. I wrote detailed comments in the original text (pdf).

My most important comments are:

add the latin name of rice to your keywords
in the Introduction section add references to literature (e.g. line 55-57)
What is the current recommended date for sowing rice in the study area?
reorder the chapters as required by the journal
provide units as required by the journal
place tables and figures after their first description in the text
correct the description of the statistics in the tables (details in pdf)

Conclusion: Is this sowing date (line 400) different from the recommended sowing dates used in the study area

correct the list of references (journal abbreviations, bold, italic, etc.)

After corrections, the manuscript may be accepted for publication in the journal Plants

Reviewer 3 Report

Dear Authors,

Manuscript: Effect of sowing date on the growth characteristics and yield of growth –constraints direct–seeding rice.

Review:

 I have a few general comments on this manuscript: this is a classical field trial work presented in this ms and will be better fitting to another journal publishing more apply research and work, e.g. Rice, Crop Res. or Agriculture.

I am very familiar with the type of rice field work and data processing and I think that 13 authors for the work presented here are just too much!

Also, this ms need language improvement and corrections in many places.

Other corrections in the text are needed:

1.       Introduction

Need improvement.

2.       Results

Lines 80 -82 need to be removed

3.       M&M

 Line 97 add the citation for the Kjeldahs method.

Line 108 starts a  new paragraph.

Line 114 fertilization management, start a new paragraph.

 Line 135 correct spacing.

Fig 1 needs to be placed into the results section! 

4.       Results

The long abbreviations of the rice genotypes used in this study are difficult for readers to follow especially in this section. Authors should think about how to make this easier and more transparent throughout the entire manuscript to follow.

Line 201 text revision is needed, redundant words.

5.       Discussion

Lines 329-337 Difficult to read and understand. Need better formulation.  

Generally can be improved.

Acknowledgment  Is redundant with Funding, please clarified these two sections.

20.3.2023